# Rapid Phenotype-Driven Gene Sequencing with the NeoSeq Panel: A Diagnostic Tool for Critically Ill Newborns with Suspected Genetic Disease

**DOI:** 10.3390/jcm9082362

**Published:** 2020-07-23

**Authors:** María José de Castro, Emiliano González-Vioque, Sofía Barbosa-Gouveia, Enrique Salguero, Segundo Rite, Olalla López-Suárez, Alejandro Pérez-Muñuzuri, María-Luz Couce

**Affiliations:** 1Diagnosis and Treatment of Congenital Metabolic Diseases Unit (UDyTEMC), Neonatology Division, Department of Pediatrics, Clinical University Hospital of Santiago de Compostela, IDIS-Health Research Institute of Santiago de Compostela, 15706 Santiago de Compostela, Spain; maria.jose.de.castro.lopez@sergas.es (M.J.d.C.); emiliano.gonzalez.vioque@sergas.es (E.G.-V.); sofia.bsg@gmail.com (S.B.-G.); olalla.elena.lopez.suarez@sergas.es (O.L.-S.); alejandro.perez.munuzuri@sergas.es (A.P.-M.); 2Centro de Investigación Biomédica en Red (CIBER)—CIBER DE ENFERMEDADES RARAS (CIBERER),, Pabellón 11, 28029 Madrid, Spain; 3MetabERN, Via Pozzuolo, 330, 33100 Udine, Italy; 4Faculty of Medicine, University of Santiago de Compostela, 15706 Santiago de Compostela, Spain; 5Neonatology Department, Malaga Regional Hospital, Malaga Biomedical Research Institute-IBIMA, 29011 Malaga, Spain; ensalgar@yahoo.es; 6Neonatology Unit, Miguel Servet University Hospital, 50009 Zaragoza, Spain; sriteg@salud.aragon.es

**Keywords:** critically ill newborn, genomic sequencing, genetic diagnosis, trio sequencing

## Abstract

New genomic sequencing techniques have shown considerable promise in the field of neonatology, increasing the diagnostic rate and reducing time to diagnosis. However, several obstacles have hindered the incorporation of this technology into routine clinical practice. We prospectively evaluated the diagnostic rate and diagnostic turnaround time achieved in newborns with suspected genetic diseases using a rapid phenotype-driven gene panel (NeoSeq) containing 1870 genes implicated in congenital malformations and neurological and metabolic disorders of early onset (<2 months of age). Of the 33 newborns recruited, a genomic diagnosis was established for 13 (39.4%) patients (median diagnostic turnaround time, 7.5 days), resulting in clinical management changes in 10 (76.9%) patients. An analysis of 12 previous prospective massive sequencing studies (whole genome (WGS), whole exome (WES), and clinical exome (CES) sequencing) in newborns admitted to neonatal intensive care units (NICUs) with suspected genetic disorders revealed a comparable median diagnostic rate (37.2%), but a higher median diagnostic turnaround time (22.3 days) than that obtained with NeoSeq. Our phenotype-driven gene panel, which is specific for genetic diseases in critically ill newborns is an affordable alternative to WGS and WES that offers comparable diagnostic efficacy, supporting its implementation as a first-tier genetic test in NICUs.

## 1. Introduction

While genetic disease is suspected in over 50% of the children who are admitted to neonatal intensive care units (NICUs) and die during the first year of life, diagnosis is confirmed in only 20%–30% of cases, often post-mortem [1,2]. Multiple factors hinder genetic diagnosis in neonates, including genetic heterogeneity, there are over 5587 known genetic diseases [3], clinical heterogeneity (e.g., the appearance of formes frustes of classical phenotypes) [4,5], and comorbidity due to the increased fragility of neonates [6,7]. Moreover, disease tends to progress faster in neonates than in other stages of life.

Technological advances in gene sequencing have enabled rapid reading of any part of the genome at an affordable price [8,9]. NICUs are a key target for the implementation of genomic tools [10,11]. Timely and specific diagnosis of newborns can have critical implications for health and wellbeing for the remainder of an infant’s life. The few pilot studies focused on the use of this technology that have been conducted in NICU settings [12,13,14,15,16,17,18,19,20,21,22,23] have yielded highly promising results. Nonetheless, further advances in this field will be required to make precision, personalized, and predictive medicine a reality. In particular, integration and interpretation of the data produced by genomic sequencing is a key obstacle to the incorporation of these strategies into clinical routine practice [24,25]. However, phenotype-based filtering and prioritization could greatly facilitate the interpretation of genetic variants detected by genome sequencing [26,27].

In this pilot study, we evaluated the utility of rapid selective gene panel trio sequencing in critically ill newborns with suspected genetic disorders. We created a specific gene panel, NeoSeq, consisting of 1870 human genes associated with neurological or metabolic disorders and congenital malformations of early onset (i.e., during the first two months of life). We hypothesized that a rapid gene panel consisting of genes of known function, with established associations with diseases of early onset, could simplify data interpretation while offering diagnostic rates comparable to those achieved using whole genome sequencing (WGS), whole exome sequencing (WES), or clinical exome sequencing (CES). Here, we describe the diagnostic rate and turnaround time achieved using this approach, the prevalence and inheritance patterns of the diagnosed diseases, and the effect of a positive molecular diagnosis on patient clinical management. Furthermore, we compare our results with those of similar prospective studies using next generation sequencing (NGS) technologies in cohorts of critically ill newborns.

## 2. Methods

### 2.1. Study Design

This multicenter prospective study included consecutive patients who were admitted to a level IIIB/C NICU in three Spanish reference hospitals during a 24 months period. Patients underwent selective genetic screening for early diagnosis. Participants for this study were recruited through the Clinical University Hospital of Santiago, Malaga Regional Hospital, and the Miguel Servet Clinical University Hospital, with the approval of the Santiago-Lugo Research Ethics Committee (2018/366). Written informed consent was obtained from the legal guardians of all participating patients upon enrolment in the study. Participants for this study were recruited through the Clinical University Hospital of Santiago de Compostela, Malaga Regional Hospital, and the Miguel Servet Clinical University Hospital, with the approval of the Santiago-Lugo Research Ethics Committee (2018/366).

### 2.2. Study Population

Critically ill patients of less than two months of age who met one or more of the following criteria were considered eligible for inclusion: (i) Congenital malformations not obviously related to a clinically identifiable genetic syndrome, (ii) metabolic decompensation not associated with biochemical parameters and/or neonatal screening indicating suspected hereditary metabolic disease, (iii) epilepsy or neurodevelopmental diseases of probable genetic origin. Individuals with clear indications of a specific syndrome that could be tested by targeted analysis of known genes or structural variations were excluded from the study.

### 2.3. Study Variables

For each patient, the following variables were evaluated: Family history, consanguinity data, maternal obstetric history, sex, anthropometric parameters, age of symptom onset, clinical signs and symptoms presented, laboratory test parameters, imaging variables and additional tests carried out to identify the underlying disease, treatments administered (pharmacological, nutritional, respiratory support, dialysis, or other invasive measures), genetic analyses of patient and parents. A list of possible differential diagnoses was generated based on the clinical findings obtained using the Phenomizer clinical diagnostics application (compbio.charite.de/phenomizer/) [28]. Upon obtaining the results of the genetic study, the clinical and molecular variables were correlated.

### 2.4. Statistical Analysis

The Chi-squared test was used to compare diagnostic rates between groups. Phenomizer was used to identify candidate diseases based on the clinical features of each patient. *p*-values were estimated by Monte Carlo random sampling and corrected for multiple testing using the Benjamini and Hochberg method. A *p*-value < 0.05 was considered statistically significant.

### 2.5. Procedures

**Blood sample collection from patient and parents.** A 3-mL sample of blood per individual from the trio (patient and parents) to be analyzed was collected in an ethylenediaminetetraacetic (EDTA) tube (BD. Franklin Lakes, NJ, US) and transported by express courier from the patient’s reference hospital at room temperature in a padded, hermetically sealed envelope.

**Genetic testing.** DNA was isolated from 400 μL of fresh blood in EDTA following standard procedures. Enrichment for focused exome analysis covering the exons and flanking introns (±10 pb) of 1870 genes associated with metabolic, neurological, and dysmorphic diseases of infants (NeoSeq, Appendix A) was carried out using the SureDesign tool (Agilent Technologies Inc., Santa Clara, CA, USA). Candidate genes were selected by a multidisciplinary team meeting (MDT), which included research bioinformatics analysts, clinical geneticists, neonatal intensivists, neurologists and pediatricians specialized in inborn errors of metabolism. Enriched libraries were sequenced on the NextSeq platform (Illumina Inc., San Diego, CA, USA) following the manufacturer recommendations to achieve an average coverage of 100X using the NextSeq 500/550 Mid Output V2 kit, (Illumina Inc., San Diego, CA, USA), (150 cycles), which provides 18 Gb of sequence data. For each NextSeq run, a single trio was sequenced.

Variant annotation and filtering were performed using pipelines developed in-house (see Appendix B for detailed description).

## 3. Results

Between the period of January 2018 and December 2019, 51 newborns were admitted to the NICU with a suspected diagnosis of genetic disease. Of the identified newborns, 10 of them were excluded due to clear indications of a specific syndrome that could be tested by targeted analysis of known genes or structural variations. A total of 41 families were approached and offered genomic analysis with NeoSeq. Of them, 37 families consented to join the study. Of the recruited patients, four newborns were not sequenced because the genetic study of the parents was not available. Finally, 33 newborns (16 females and 17 males) met our inclusion criteria and were included in the study (Appendix A). The median age at inclusion was 26 days (range, four days to two months). Two patients died before conclusion of the study, one prior to diagnosis and the other after diagnosis was established and palliative care initiated. For all patients, DNA was obtained from both parents. All probands were phenotyped using human phenotype ontology (HPO) terms extracted manually from electronic health records. The median number of HPO terms per proband was four, and 58% of patients had at least four terms. Based on the predominant presenting clinical symptoms, patients were classified into the following groups: Congenital anomalies (n = 8), neurological symptoms, including encephalopathy, developmental regression, seizures, and hypotonia (n = 17), suspected metabolic diseases (n = 7), and severe intrauterine growth retardation (n = 1). We observed no significant differences in rates of diagnosis between groups (*p* = 0.572). For each of the 33 patients included in this study, demographic data, clinical presentation, and standardized phenotypes using HPO terms are shown in Appendix A. Median diagnostic turnaround time was 7.5 days (range, 4–11 days).

A genetic diagnosis was established for 39.4% (n = 13) of our 33 patients (Table 1). Analysis of inheritance patterns revealed compound heterozygous variants in six patients for the following genes (associated autosomal recessive disorders are shown in parentheses): GFM1 (combined oxidative phosphorylation deficiency 1), SUCLA2 (mitochondrial DNA depletion syndrome 5), PNPT1 (combined oxidative phosphorylation deficiency 13), CPS1 (Carbamoyl-phosphate synthetase 1[CPS1] deficiency), COQ4 (primary coenzyme Q10 deficiency), and ERBB3 (lethal congenital contracture syndrome type 3). We identified seven patients with heterozygous variants in the following genes (associated autosomal dominant disorders are shown in parentheses): KCNQ2 (early infantile epileptic encephalopathy 7, three patients), CHD2 (nemaline myopathy), COL4A1 (cerebral small vessel disease), and SOX10 (Waardenburg syndrome-Hirschsprung’s disease). Of these seven patients, five carried de novo variants and two had inherited the variant from a single parent, who presented a less severe phenotype. Only in 38.4% of cases was the phenotype of the newborn predicted based on the molecular diagnosis returned by Phenomizer (*p* < 0.05).

Likely pathogenic variants were identified in three patients but were ruled out owing to their presence in a healthy parent. These cases are presented in detail in Table 2.

Diagnoses of Prader–Willi syndrome and Steinert dystrophy were not detected in patients #20 and #21, respectively, but were established based on specific tests performed later in life as the respective diseases progressed. Patient #20 carried a deletion in the paternal allele in 15q11-q13 and patient #21 harbored a pathological expansion of 2333 CTG copies in the DMPK gene.

We next evaluated the impact of genetic diagnosis on four distinct aspects of medical management for a period of three months: (a) Redirection of care (towards withdrawal of intensive care or initiation of palliative care), (b) initiation of new subspecialist care, including additional extension/follow up studies, and (c) changes in medication or diet. We found that establishing a genomic diagnosis directly affected medical management in 12 of the 13 patients (Table 3), resulting in changes in medication or diet (10 patients), initiation of new subspecialist care (six patients), or withdrawal of intensive care treatment /initiation of palliative care (six patients). Moreover, all patients and families benefited from timely genetic counseling based on a concrete diagnosis.

Finally, we compared our results with those of previously published in prospective studies in which massive sequencing techniques (WGS, WES, CES) were applied to newborns admitted to NICUs with a suspected genetic disorder (Table 4). Our comparison included 12 studies published between 2016 and 2020. Nine were cohort studies, and three were randomized clinical trials. The selected studies applied the following sequencing techniques: CES (three studies) [12,13,17], WES (four studies) [16,18,19,23], WGS with subsequent filtering by CES (1 study) [14], WGS (three studies) [15,20,22], and WES + WGS (one study) [21]. The total number of patients included was 1073. The median diagnostic rate was 37.2% (range, 13.2%–58%). After adjusting for the NGS technology employed, the following diagnostic rates were obtained: CES, 43.2% (range, 32.4%–50.8%), WES, 40.4% (15.6%–56%), WGS with subsequent filtering by CES, 30.4%, and WGS, 32.8% (range, 13.2%–47.7%). The median diagnostic turnaround time was 22.3 days.

## 4. Discussion

Genomic sequencing has emerged as one of the most promising diagnostic tools in the field of neonatology. Studies have demonstrated that the adoption of this technology as a first-tier test for genetic diagnosis of severely ill newborns increases the diagnostic rate and reduces the time to diagnosis, improving outcomes while reducing health costs [29,30]. In the present study, we evaluated the diagnostic efficacy and clinical impact of NeoSeq, a gene panel that consists of 1870 genes and is specifically designed to facilitate the diagnosis of critically ill newborns with suspected genetic disorders. In total, 33 newborns and their parents underwent trio sequencing using the NeoSeq panel. Newborns with neurological symptoms comprised the largest group (n = 17), followed by those with multiple congenital anomalies (n = 8) and suspected metabolic diseases (n = 7). These phenotypic groupings corresponded to those previously described in other studies that used rapid NGS strategies to diagnose NICU patients [12,23].

The resulting diagnostic rate (13/33 patients [39.4%]) was comparable to that previously described for more extensive trio or singleton sequencing approaches, such as WES [16,21] and WGS [14,15,21,22], and even higher than that reported for WGS in NICU patients [20]. While higher diagnostic rates should be expected from WGS than WES, and from WES than CES or targeted panels, we found that the diagnostic rate does not correlate with the size of genome portion interrogated (Figure 1): NeoSeq provided diagnostic rates similar to those obtained with WGS and WES approaches, despite interrogating 10 times (WES) and 500 times (WGS) fewer bases.

Two factors may help explain these discrepancies. First, variants are usually excluded if they have no predicted or known functional consequences [31,32,33,34]. This limits the analysis to exonic or intronic variants in known disease-associated genes or variants with demonstrated pathogenicity located in noncoding regions. Consequently, the phenotype of the newborn guides the analysis to a small group of candidate genes. Therefore, regardless of how extensive the genomic analysis, genetic diagnosis is limited to a shortlist of genes [14,21] and variant types [22]. Our results, and those of others using a similar approach [17], demonstrate that a simpler approach, based in the analysis of panels of genes with known functions and disease associations, can be a useful and more cost-effective alternative to WGS or WES approaches, yielding comparable diagnostic rates. The second factor to consider is the association between the diagnostic rate and the severity of the clinical phenotype of the study participants. All patients recruited in our study were critically ill newborns that required intensive care. In two previous studies in which patients with severe and less severe disease were assigned to distinct cohorts, better outcomes were observed for the severely ill group in terms of diagnostic rate and the impact on clinical management. In the NSIGHT2 trial [21], patients were randomized into two groups, which were analyzed using rapid WES (rWES) or rapid WGS (rWGS), except for severely ill patients, who were screened using ultra-rapid WGS (urWGS). The resulting diagnostic rate was 46% in the urWGS group, as compared with 19% and 20% in the rWGS in rWES groups, respectively. Meng et al. [13] used WES to perform proband exome, trio exome, and critical trio exome sequencing (a rapid genomic assay for seriously ill infants) in three groups of infants within the first 100 days of life. The diagnostic rate in the critical trio exome group was higher than that obtained for the proband and trio exome groups. Moreover, molecular diagnoses directly affected medical management in 76.9% of patients in the critical trio exome group, as compared with 42.9% in the other groups. These results suggest that a diagnosis strategy based on the sequencing of a panel of genes associated with severe neonatal phenotypes yields better outcomes because those patients likely have a genetic disorder and can greatly benefit from accurate and early diagnosis.

In our study, we applied a family trio analysis approach (testing newborns and both parents), as this is an effective strategy to manage the wealth of genetic variants identified by NGS. This approach can be used to easily identify de novo variants, filter out rare benign familial variants, and establish inheritance patterns in recessive disorders [18,34]. We identified five de novo causative variants among our patients and, most importantly, in three cases, we ruled out potential phenotype-associated variants after detecting them in a single asymptomatic parent (see Table 2). In our cohort, two patients for whom NeoSeq failed to establish a molecular diagnosis were later diagnosed with Prader–Willi syndrome and Steinert dystrophy, respectively. This is a common limitation of current NGS technologies, including WGS and WES, which lack the capacity to identify repetitive sequences, homologous genes, and epigenetic modifications [35,36].

In addition to the diagnostic rate of 39.4% (13/33 patients), the rapid molecular diagnosis achieved with NeoSeq (mean diagnostic turnaround time, 7.5 days) impacted medical management in 92.3% of patients, as reflected in changes in medication or diet (10/13 patients), initiation of new subspecialist care (6/13 patients), or withdrawal of intensive care treatment/redirection to palliative care (6/13 patients). In two patients, NeoSeq enabled diagnosis of inborn errors of metabolism: CPS1 deficiency and primary coenzyme Q10 deficiency, allowing the early initiation of ammonia scavenger therapy combined with a protein-restricted diet [37] and coenzyme Q10 supplementation [38], respectively. In CPS1 deficiency treatment can prevent hyperammonemic crisis and adverse neurological outcomes, while in primary forms of coenzyme Q10 deficiency treatment can prevent the progression of both steroid-resistant nephrotic syndrome and encephalopathy, hence the critical importance of a prompt diagnosis. In three other patients who presented primarily with epilepsy, gene sequencing identified de novo *KCNQ2* variants, for which anticonvulsant treatment with sodium channel blockers is highly recommended [39] and reduces the neurodevelopmental impairment associated with the disease. Although some of the established diagnoses did not lead to institution of effective treatments, they provided important information regarding prognosis and disease management, putting an end to a potentially lengthy diagnostic odyssey. Furthermore, a genetic diagnosis allows for testing of other at-risk family members and provision of reproductive counseling where necessary [24,40].

Analysis of diagnostic rate according to phenotype shows that neonates with neurological symptoms and suspected inborn errors of metabolism benefited most from the NeoSeq panel (diagnostic yield, 47.1% and 42.9%, respectively). By contrast, only in two patients with congenital anomalies was a definitive causative variant identified (28.6%). Phenomizer predicted a genetic diagnosis based on phenotype with statistical significance (*p* < 0.05) in only five patients. This rate of diagnosis prediction is similar to that reported by Brunelli et al. [17] using a targeted gene panel (40%), and higher than that reported by French et al. [20] (10%) using WGS. Based on their data, French et al. proposed agnostic analysis of genomic data, as opposed to phenotype-driven analysis. However, analyses using WGS or WES cannot be completely agnostic owing to the aforementioned limitations affecting variant interpretation.

## 5. Conclusions

In summary, rapid trio sequencing with our phenotype-driven gene panel specific for genetic diseases in critically ill newborns is an affordable alternative to WGS and WES that offers comparable diagnostic efficacy, supporting its implementation in routine clinical practice as a first-tier genetic test in NICUs.

## Figures and Tables

**Figure 1 jcm-09-02362-f001:**
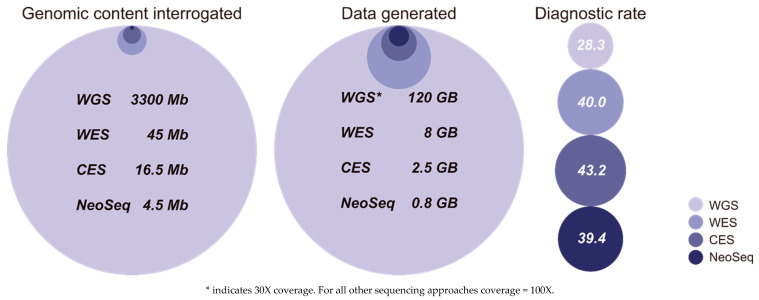
Schematic comparing NeoSeq and other diagnostic genome sequencing strategies (WGS^14,15,20,22^, WES^16,18,19,23,^ and CES^12,13,17^) showing the size of the genomic portion interrogated, the amount of data generated, and the median diagnostic rates achieved for each approach. To facilitate visual comparison, data are represented as the relative area of each circle. For WGS and WES, information on the genomic portion interrogated and data generated data were extracted from www.illumina.com/wes-wgs. For CES, information on the genomic portion interrogated and data generated was estimated for 6000 genes. Diagnostic rates were calculated as the mean of the results of the cited studies. Total number of patients for each genome sequencing strategy: CES, n = 306; WES, n = 242; WGS, n = 395.

**Table 1 jcm-09-02362-t001:** Demographic and clinical data and genetic findings in patients with definitive/probable causative variants.

Cases	Age	Sex	HPO	TAT	Gene	Variants	Phenomizer	Disorder	MIM	Parents/Inheritance
1	25 d	M	Intrauterine growth retardation Neonatal respiratory distress Congenital microcephaly Unilateral cryptorchidism Neonatal hypoglycemia Chronic metabolic acidosis Lactic acidosis	7 d	*GFM1*	c.[640A>T];[ 1199G>C] (p.[Ile214Phe];[Arg400Pro])	*p* = 0.0343	Combined oxidative phosphorylation deficiency 1	#609060	Carriers Autosomal recessive
2	1 m 15 d	M	Methylmalonic aciduria Abnormality of myocardium Chronic metabolic acidosis Lactic acidosis	7 d	*SUCLA2*	c.[850C>T];[ 850C>T] (p.[Arg284Cys];[Arg284Cys])	*p* = 0.0271	Mitochondrial DNA depletion syndrome 5	#612073	Carriers Autosomal recessive
3	2 m	F	Failure to thrive Weight loss Irritability Infantile axial hypotonia Developmental regression Hyperglycinemia	4 d	*PNPT1*	c.[1495+55_1495+80del]; [1495+55_1495+80del]	n.s. *p* = 0.119	Combined oxidative phosphorylation deficiency 13	#614932	Carriers Autosomal recessive
4	20 d	F	Seizures Epileptic encephalopathy	6 d	*KCNQ2*	c.778C>T (p.His260Tyr)	*p* = 0.04	Epileptic encephalopathy, early infantile 7	#613720	De novo Autosomal dominant
5	18 d	M	Seizures Epileptic encephalopathy	7 d	*KCNQ2*	c.1016 T>A (p.Leu339Gln)	*p* = 0.04	Epileptic encephalopathy, early infantile 7	#613720	Father, epilepsy in infancy Autosomal dominant
6	25 d	F	Seizures Epileptic encephalopathy	6 d	*CDKL5*	c.616G>T (p.Asp206Tyr)	n.s. *p* = 0.1386	Epileptic encephalopathy, early infantile 2	#300672	Mother, epilepsy Autosomal dominant X-linked
7	6 d	M	Neonatal respiratory distress Seizures Epileptic encephalopathy Chronic metabolic acidosis	7 d	*KCNQ2*	c.1658G>A (p.Arg553Gln)	n.s *p* = 0.471	Familiar neonatal seizures 1	#613720	De novo Autosomal dominant
8	10 d	M	Vomiting Hyperammonemia Hypoketotic hypoglycemia Fetal pyelectasis Thyroid-stimulating hormone excess	8 d	*CPS1*	c.[1201G>C];[2810T>A] (p.[Gly401Arg];[Ile937Asn])	n.s. *p* = 0.0624	CPS1 deficiency	#608307	Carriers Autosomal recessive
9	1 m	M	Generalized neonatal hypotonia Seizures Lethargy Cerebellar atrophy	10 d	*COQ4*	c.[202G>C];[718C>T] (p.[Asp68His];[Arg240Cys])	n.d.	Primary coenzyme Q10 deficiency 7	#616276	Carriers Autosomal recessive
10	3 d	F	Premature birth Respiratory insufficiency due to muscle weakness Generalized hypotonia, Encephalopathy	10 d	*ACTA1*	c.614C>A (p.Thr205Lys)	n.s. *p* = 0.582	Nemaline myopathy 3	#161800	De novo Autosomal dominant
11	9 d	F	Leukoencephalopathy Cataracts Renal cysts Bleeding digestive	5 d	*COL4A1*	c.2906 G>A (p.Gly969Glu)	n.s. *p* = 0.5810	Small vessel vascular brain disease	#175780	De novo Autosomal dominant
12	12 d	M	Joint contractures Microretrognathia Microtia Micropenis Hypotonia Hearing loss	6 d	*ERBB3*	c.[1184-9A>G];[1184-9A>G]	n.s. *p* = 0.4243	Lethal contracture syndrome type 2	#607598	Carriers Autosomal recessive
13	15 d	F	Hirschsprung disease Congenital hearing loss Opsoclonus	7 d	*SOX10*	c.850G>T (p.Glu284Ter)	*p* = 0.0323	Waardenburg–Shah syndrome	#142623	De novo Autosomal dominant

Abbreviations: d, day; HPO, human phenotype ontology; F, female; m, month; M, male; MIM, Mendelian Inheritance in Man; n.d., not detected; n.s, not significant; TAT, turnaround time to diagnosis.

**Table 2 jcm-09-02362-t002:** Demographic and clinical data and genetic findings in patients with non-definitive/non probable causative variants.

Cases	Age	Sex	HPO	TAT	Gene	Mutations	Phenomizer	Disorder	MIM	Parents/Inheritance
14	22 d	F	Preeclampsia Neonatal respiratory distress Hypoglycemia	7 d	*MTOR*	c.126G>T (p.Lys42Asn)	n.d.	-Focal cortical dysplasia type II - Smith-Kingsmore Syndrome	#607341 #616638	Father asymptomatic carrier
15	29 d	F	Intrauterine growth retardation Meconium ileus Neonatal respiratory distress Generalized neonatal hypotonia Congenital hip dislocation 10 pairs of ribs	10 d	*NALCN*	c.2507C>G (p.Pro836Arg)	n.d.	Congenital contractures of the limbs and face, hypotonia and developmental delay	#616266	Father asymptomatic carrier
16	4 d	M	Focal seizures	9 d	*TSC2*	c.1724T>C (p.Leu575Pro)	n.d.	Tuberous sclerosis complex	#613254	Father asymptomatic carrier

d, day; F, female; HPO, human phenotype ontology; M, male; MIM, Mendelian Inheritance in Man; n.d., not detected; TAT, turnaround time.

**Table 3 jcm-09-02362-t003:** Impact of molecular diagnosis on medical management.

Cases	Disorder	Changes in Medication or Diet	Initiation of New Subspecialist Care	Withdrawal of Intensive Care Treatment/Initiation of Palliative Care	Genetic Counseling
1	Combined oxidative phosphorylation deficiency 1	Mitochondrial cocktail Avoid valproate	Neurologist Hepatologist Audiologist Ophthalmologist	Yes	Yes
2	Mitochondrial DNA depletion syndrome 5	Mitochondrial cocktail	Neurologist Audiologist Ophthalmologist	Yes	Yes
3	Combined oxidative phosphorylation deficiency 13	Mitochondrial cocktail Avoid valproate	Cardiologist Audiologist Ophthalmologist	Yes	Yes
4	Epileptic encephalopathy, early infantile 7	Sodium channel blocker (phenytoin) Avoid retigabine and ezogabine	No	No	Yes
5	Epileptic encephalopathy, early infantile 7	Sodium channel blocker (carbamazepine) Avoid retigabine and ezogabine	No	No	Yes
6	Epileptic encephalopathy, early infantile 2	Ketogenic diet	Gastroenterologist	No	Yes
7	Familiar neonatal seizures 1	Sodium channel blocker (phenobarbital) Avoid retigabine and ezogabine	No	No	Yes
8	CPS1 deficiency	Protein restricted diet Arginine/citrulline	Specialist in inborn errors of metabolism	No	Yes
9	Primary coenzyme Q10 deficiency 7	CoQ10	Cardiologist Ophthalmologist Audiologist	No	Yes
10	Nemaline myopathy	Pyridostigmine Carnitine	No	Yes	Yes
11	Small vessel brain disease	- (exitus)	- (exitus)	- (exitus)	Yes
12	Lethal contracture syndrome type 2	No	No	Yes	Yes
13	Waardenburg–Shah syndrome	No	No	Yes (exitus)	Yes

**Table 4 jcm-09-02362-t004:** Previous prospective studies of the diagnostic performance of WES, WGS, and CES in newborns with a suspected genetic disorder.

Reference	Date	Study Type	Sequencing Type	Study Population	Rate of Diagnosis	Diagnostic Turnaround Time
Daoud, et al. [12]	2016	Cohort	CES	n = 8	4/8 (50%)	15.2 w
Meng, et al. [13]	2017	Cohort	CES Trio CES Critical trio CES	n = 178 n = 37 n = 63	58/178 (33%) 12/37 (32%) 32/63 (51%)	95 d 51 d 13 d
van Diemen, et al. [14]	2017	Cohort	WGS filtered by CES	n = 23	7/23 (30%)	12 d
Petrikin, et al. [15]	2018	RCT	Trio rWGS	n = 64	21/64 (33%)	13 d
Stark, et al. [16]	2018	Cohort	rWES	n = 40	21/40 (52%)	16 d
Brunelli, et al. [17]	2019	Cohort	rCES	n = 20	10/20 (50%)	9.6 d
Ceynah-Birsoy, et al. [18]	2019	Cohort	WES	n = 32	5/32 (16%)	-
Elliot, et al. [19]	2019	Cohort	WES	n = 25	14/25 (56%)	7.2 d
French, et al. [20]	2019	Cohort	Trio WGS	n = 106	14/106 (13%)	21 d
Kingsmore, et al. [21]	2019	RCT	rWGS rWES urWGS	n = 94 n = 95 n = 24	18/94 (19%) 19/95 (20%) 11/24 (46%)	11 d 11.2 d 4.6 d
Wang, et al. [22]	2020	RCT	Trio WGS	n = 84	32/84 (38%)	4 d
Gubbels, et al. [23]	2020	Cohort	Trio WES	n = 50	29/50 (56%)	4.9 d

Abbreviations: CES, clinical exome sequencing; RCT, randomized clinical trial; r, rapid; ur, ultrarapid; WES, whole exome sequencing; WGS, whole genome sequencing.

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
