# Peer review of "Rapid Phenotype-Driven Gene Sequencing with the NeoSeq Panel: A Diagnostic Tool for Critically Ill Newborns with Suspected Genetic Disease"

_jcm, 2020, doi:10.3390/jcm9082362_

Round 1

Reviewer 1 Report

Dr de Castro and colleagues present a prospective study of genomic sequencing in the NICU.  They identify a cohort of 33 neonates and sequence them  (and their parents) with a comprehensive panel of disease genes that present in this time period.  The panel is similar to the neonatal crisis panel offered by Prevention Genetics (1757 genes).   They see a diagnostic rate of 13/33 and were able to see the clinical benefits of a molecular diagnosis in almost all.

Overall this is a well written and well considered study. Key was the use of strict inclusion criteria and the use of HPO terms. The conclusion is balanced and results placed in the appropriate context.  My concerns/comments are minor.

Micorarray. I am curious about microarrays for these newborns.  Were all the individuals tested with a microarray and found to be negative prior to inclusion in the study? In particular the newborns with congenital malformations.

The variants need to be interpreted with ACMG criteria and presented in Table 1.  Mutation should be referred to as variant.

Very minor comment but can the authors edit the second sentence of the first paragraph? The sentence has three bracketed texts that is very distracting.  It can be rewritten without the brackets as it really disrupts flow of sentence.

Not for the review but in the future, perhaps the authors can optimize blood collection to reduce the total volume (they collect 3ml and used 400ul) in these fragile neonates

Author Response

Overall this is a well written and well considered study. Key was the use of strict inclusion criteria and the use of HPO terms. The conclusion is balanced and results placed in the appropriate context.  My concerns/comments are minor.

  1. Micorarray. I am curious about microarrays for these newborns.  Were all the individuals tested with a microarray and found to be negative prior to inclusion in the study? In particular the newborns with congenital malformations. ANSWER: Thank you very much for your appropriate question. Indeed, in our protocol we have asked for the karyotype when a congenital malformation is suspected but the results are given in a minimum of 15 days. Therefore, we have the results of the genetic analysis from NeoSeq information before the kariotype.
  2. The variants need to be interpreted with ACMG criteria and presented in Table 1.  Mutation should be referred to as variant. ANSWER: We agree and we have changed accordingly.
  3. Very minor comment but can the authors edit the second sentence of the first paragraph? The sentence has three bracketed texts that is very distracting.  It can be rewritten without the brackets as it really disrupts flow of sentence. ANSWER: We agree and we have changed accordingly.

Reviewer 2 Report

In a prospective study, the authors investigated how rapidly and in how many cases an early initiated next generation sequencing diagnostic procedure can lead to a reliable diagnosis in critically ill newborns and what consequences result from this for the treatment and clinical management of the patients. It turned out that in 13 of 33 recruited newborns (39.4%) a genetic diagnosis could be made, with a mean turnaround time of 7.5 days, and a consequence for clinical management in the majority of cases (in 10/13 patients, 76.9%). Sequencing was carried out as a trio approach including parental data sets. Interestingly, the study was not conducted as whole exome (WES) or whole genome (WGS) sequencing, but sequencing was limited to a physical panel of 1870 genes known to be associated with severe early onset disorders. Despite this, a similar diagnostic rate was achieved compared to previous studies in which a WES or WGS approach was used. The data of this carefully conducted and documented study are very important because in many countries the early use of NGS testing is being discussed and must be justified before health authorities.

Comments

- In order to make the results comparable with future studies in which phenotype-based panels are used, the list of genes under investigation (“NeoSeq”) should be provided.

-  table 1: Variants should be also described at protein level  (p.-position)

Minor comments:

Abstract line 28: “…, but a lower median diagnostic…”, probably should read “but a higher median diagnostic…”

Abstract, line 31: “its implementation in as a first-tier genetic test”, “in” should be omitted

Author Response

The data of this carefully conducted and documented study are very important because in many countries the early use of NGS testing is being discussed and must be justified before health authorities.

  1. In order to make the results comparable with future studies in which phenotype-based panels are used, the list of genes under investigation (“NeoSeq”) should be provided. ANSWER: The complete list of genes is added as Table S1 in supplementary material.
  2. Table 1: Variants should be also described at protein level  (p.-position). ANSWER: We added the protein level in Table 1
  3. Minor comments: Abstract line 28: “…, but a lower median diagnostic…”, probably should read “but a higher median diagnostic…”. ANSWER: You are right and we have corrected this in the manuscript
  4. Abstract, line 31: “its implementation in as a first-tier genetic test”, “in” should be omitted. ANSWER: We agree and we have corrected
